# Influenza Vaccination and Health Outcomes in COVID-19 Patients: A Retrospective Cohort Study

**DOI:** 10.3390/vaccines9040358

**Published:** 2021-04-08

**Authors:** Pasquale Domenico Pedote, Stefano Termite, Andrea Gigliobianco, Pier Luigi Lopalco, Francesco Paolo Bianchi

**Affiliations:** 1Epidemiology Unit, Brindisi Local Health Authority, 72015 Fasano, Italy; pasquale.pedote@asl.brindisi.it; 2Public Health Unit, Brindisi Local Health Authority, 72100 Brindisi, Italy; stefano.termite@asl.brindisi.it; 3Health Management, Brindisi Local Health Authority, 72100 Brindisi, Italy; andrea.gigliobianco@asl.brindisi.it; 4Department of Health, 70100 Apulia, Italy; pierluigi.lopalco@unipi.it; 5Department of Biomedical Science and Human Oncology, University of Bari, 70121 Bari, Italy

**Keywords:** COVID-19, influenza vaccine, prevention, cause-effect relationship

## Abstract

COVID-19 is an infectious disease caused by the novel coronavirus SARS-CoV-2. Several measures aimed at containing the spread of this virus have been recommended by international and nation public health institutions, but whether the influenza vaccine, while not protective against COVID-19, nonetheless reduces disease severity is unclear. This study evaluated the potential role of influenza vaccine in reducing the rate of hospitalization and death in COVID-19 patients. COVID-19 cases recorded in the province of Brindisi (Apulia, Southern Italy) during the first pandemic wave (February–May 2020) and occurring in patients vaccinated with the influenza vaccine during the 2019–2020 influenza season were considered. From February 2020 to May 2020, 3872 inhabitants of the province of Brindisi underwent SARS-CoV-2 PCR testing and 664 (8.7%) tested positive. A multivariate analysis showed that among COVID-19 patients neither hospitalization nor death was significantly associated with influenza vaccination (*p* > 0.05), whereas within this group male sex, older age, and chronic diseases were identified as risk factors for morbidity and mortality. Our study did not show an association between the influenza vaccine and complications of COVID-19. Nonetheless, influenza vaccination must be promoted as a central public health measure, because by reducing the burden on hospitals it can greatly benefit the management of COVID-19 patients.

## 1. Introduction

COVID-19 is the infectious disease caused by the novel coronavirus SARS-CoV-2. Neither the virus nor the potentially fatal disease it causes was known before the outbreak began in Wuhan, China in December 2019. COVID-19 is now a pandemic of global proportions [1]. The World Health Organization (WHO, Geneva, Switzerland) has reported ~98,000,000 confirmed cases of COVID-19 globally, including more than 2,000,000 deaths [2]. According to the European Centre for Disease Prevention and Control, there have been 17,906,888 cases and 425,618 deaths in the EU/EEA [3]. Italy ranks first in Europe in terms of the number of COVID-19-related deaths (83,872; fatality rate: 3.4%) and the second per absolute number of cases (*n* = 2,447,443) [4]; all data are current as of 25 January 2021. The median age of patients in Italy is 48 years, with a higher proportion of cases among females (51.6%). COVID-19 patients 60+ years of age have the most severe clinical status and the highest death rates [4].

The anti-COVID-19 mass vaccination campaign started in Italy and other European countries on 27 December 2020, but its effects will not be appreciable before several months. Until then, the Center for Diseases Control and Prevention (CDC) continues to recommend the preventative measures drawn up during the initial phase of the pandemic: frequent handwashing with soap and water, avoiding close contact, covering the mouth and nose with a cloth face cover, covering coughs and sneezes, daily cleaning and disinfection of frequently touched surfaces, and daily health monitoring [5]. In addition, the CDC reported beneficial effects of influenza vaccination during the pandemic. Specifically, while the vaccine does not protect against COVID-19 [5], by reducing the risk of influenza and therefore hospitalization, vaccination can allow healthcare resources to be devoted to the care of patients with COVID-19.

Nonetheless, few studies have evaluated the effects of influenza vaccine on COVID-19 and their results have mostly been conflicting. A 2020 case-control study [6] examined the role of influenza vaccine in 261 healthcare workers, including 180 with a history of COVID-19 and 181 healthy controls. In the univariate analysis, the odds ratio of being vaccinated was 0.04 (95% confidence interval [CI] = 0.01–0.14). The authors concluded that influenza vaccine may have a protective effect in COVID-19. A 2020 US study [7] showed an inverse association of greater influenza vaccination coverage in the elderly and mortality from COVID-19, suggesting a protective effect of the influenza vaccine. However, an Italian study [8] of a sample of healthcare workers found no evidence of a relationship between the influenza vaccine and either a COVID-19 diagnosis or a positive SARS-CoV-2 serology test in this group. Furthermore, Wehenkel C [9] reported a positive correlation between COVID-19 deaths and influenza vaccination in people ≥65 years-old, although the journal’s editors reminded readers that correlation does not necessarily imply causation. By contrast, a 2020 non-systematic review of the published, pre-print, and gray literature [10] concluded that high rates of coverage by the influenza and 23-valent pneumococcal polysaccharide vaccines has the potential to not only reduce the burden of the targeted diseases but to also prevent a proportion of COVID-19 morbidity and mortality.

Thus, in this study we evaluated the impact of the influenza vaccine administered during the 2019–2020 influenza season on hospitalization and death in COVID-19 patients during the first wave of the pandemic (February–May 2020). Our study was carried out in the province of Brindisi (Apulia, South Italy, around 400,000 inhabitants), where, in February 2020, 7962 confirmed cases of COVID-19 and 172 related deaths were reported [11]. Influenza vaccine coverage in the total population during 2019–2020 reached 16.8%, and 48.2% in the age group 65 years and older.

## 2. Materials and Methods

A retrospective cohort study design was used by the Epidemiology Unit of the Department of Health of the province of Brindisi.

The surveillance platform GIAVA COVID-19 (Sincon, Apulia, Italy) was developed on the basis of the WHO Go.Data outbreak investigation tool [12] and was set up to manage the pandemic emergency in Apulia. This platform is used to record COVID-19 patients and their contacts, to collect patients’ demographic, laboratory and clinical data, the results of SARS-CoV-2 PCR (Polymerase Chain Reaction) testing, and to follow-up COVID-19 patients over the course of the disease, with an update of their health status (clinical symptoms, hospitalization, death, recovery).

For this study, data relevant to the COVID-19 cases recorded in the province of Brindisi during the first wave of the pandemic (February–May 2020) were extracted from GIAVA COVID-19 as follows: unique personal identification number (PIN), sex, age, chronic disease (YES/NO), hospitalization (YES/NO) and death (YES/NO). A COVID-19 case was defined according to international guidelines [13].

Based on the assumption of a sub-optimal compilation of data by GIAVA COVID-19, information on chronic diseases and hospitalizations was checked using the Edotto platform (Exprivia, Apulia, Italy) of the Apulian Health Information System. Edotto, set up in 2012, allows the integration of various branches of the Italian healthcare system (Health Department, Regional Health Agency, healthcare companies, general practitioners, pharmacies, hospital physicians, etc.). From Edotto’s regional archive of hospital discharge forms (SDO) [14], data on all hospitalizations of the inhabitants of the province of Brindisi from February to May 2020 were extracted, considering all records that included a reference to COVID-19, whether as the primary or the secondary diagnosis, based on the ICD-9 (International Classification of Diseases) codes as defined by the Italian Ministry of Health [15]. High-risk patients were defined according to the findings of a 2018 Italian paper by Martinelli D et al. Those authors reviewed published recommendations on medical conditions for which vaccination against influenza and other vaccine-preventable infectious diseases was indicated and then associated them with the Italian user-fee exemption codes [16]. Thus, in our study, through a specific function of Edotto, all inhabitants of the province of Brindisi for which fee exemption codes had been assigned during their lifetime were selected. To ensure the reliability of the data, Edotto was also used to identify the pharmaceutical prescriptions of the same population from August 2019 to February 2020, with codes then assigned using the Anatomical Therapeutic Chemical Classification System [17]. Finally, these datasets were decoded using the risk categories defined by Martinelli D et al. [16] and used to identify patients with at least one chronic disease. The information obtained by Edotto was matched with that extracted from the GIAVA COVID-19 platform by using the patients’ unique Personal Identification Numbers (PINs). Data relating to the hospitalizations and chronic diseases of COVID-19 patients were double checked and reviewed, when necessary.

The overall vaccination status of COVID-19 patients was assessed using the Regional Immunization Database (GIAVA). GIAVA is a computerized vaccination registry containing information on the vaccination history of every Apulian inhabitant; it can also be used to generate an immunization schedule. For the 2019–2020 influenza season, the Apulian Government recommended vaccination with the inactivated tetravalent formulation for all inhabitants between the ages of 6 months and 64 years and for those between the ages of 65 and 74 years who were in good health; for inhabitants 65–74 years of age and suffering from chronic disease and in all inhabitants ≥75 years, the adjuvanted trivalent formulation was recommended [18]. Records of influenza vaccination between October 2019 and January 2020 were extracted and matched with the data obtained from the GIAVA COVID-19 platform using the patients’ unique PINs (Figure 1).

The final dataset was created as an Excel spreadsheet that included information on sex, age at COVID-19 diagnosis, chronic disease (YES/NO), hospitalization (YES/NO), death (YES/NO), and influenza vaccination (YES/NO) and the vaccine formulation (inactivated tetravalent or adjuvanted trivalent). An anonymized data analysis was performed using the STATA MP16 software (StataCorp, College Station, TX, USA).

Continuous variables are presented as the median, interquartile range (IQR), and range, and categorical variables are presented as proportions. Cumulative incidence and the mortality rate were calculated using the province of Brindisi population, as reported in the ISTAT demographic archives, as the denominator. The age at COVID-19 diagnosis was classified in one of seven age classes (<29, 30–39, 40–49, 50–59, 60–69, 70–79, 80+), with the proportion of hospitalized patients and lethality stratified per age class.

The determinants of hospitalization (YES/NO) and death (YES/NO) were assessed using two multivariate logistic regression models (one for each outcome), considering sex (male vs. female), age at diagnosis (years), chronic disease (YES/NO), and influenza vaccination (YES/NO) as determinants. The adjusted odds ratio (aOR) was calculated together with the 95% CI. The analyses were then repeated considering only patients aged 65+ years. The Hosmer–Lemeshow test was used to assess the goodness-of-fit of the multivariate models. For all the tests, a two-sided *p*-value < 0.05 was considered to indicate statistical significance.

## 3. Results

From February 2020 to May 2020, 3872 inhabitants of the province of Brindisi were tested for SARS-CoV-2 using a PCR test and 664 (8.7%; 52.1% female) were determined to be positive. The cumulative incidence was equal to 16.95 cases per 10,000 inhabitants. The median age of the COVID-19 patients was 55 years (IQR: 39–71; range: 0–100). Distribution by age group was as follows: <29 years: *n* = 89 (13.4%), 30–39 years: *n* = 77 (11.6%), 40–49 years: *n* = 94 (14.2%), 50–59 years: *n* = 119 (18.0%), 60–69 years: *n* = 105 (15.9%), 70–79 years: *n* = 67 (10.1%), and 80+ years: *n* = 111 (16.8%).

At least one chronic disease was recorded in 315 (47.6%) patients, with a higher proportion among the population 70+ years. Among the 152 (23.0%) patients requiring hospitalization, the proportion increased with increasing age (Figure 1). Sixty-five deaths were recorded (fatality rate: 9.8%; mortality: 1.66 per 10,000 inhabitants), with a higher rate in patients 80+ years of age (Figure 2).

According to the patients’ immunization status, downloaded from GIAVA, 190 of the 662 (28.7%) patients were vaccinated against influenza; of these, 130/190 (68.4%) received the adjuvanted trivalent influenza vaccine and 60/190 (31.6%) received the inactivated tetravalent influenza vaccine. Vaccine coverage in patients 65+ years was 63.1% (*n* = 142/225), of which 130/142 (91.6%) received the adjuvanted trivalent influenza vaccine and 12/142 (8.4%) received the inactivated tetravalent influenza vaccine.

The multivariate analysis showed a statistically significant association between hospitalization and sex (aOR = 2.3; 95%CI = 1.5–3.5), age (aOR = 1.03; 95%CI = 1.02–1.05), and chronic disease (aOR = 2.7; 95%CI = 1.6–4.4). When the same analysis was performed considering only patients 65+ years of age, hospitalization was associated with chronic disease (aOR = 6.0; 95%CI = 2.7–13.1). In neither analysis was influenza vaccination associated with the outcome (*p* > 0.05; Table 1), regardless of the vaccine formulation (*p* > 0.05; not shown).

The multivariate analysis showed a statistically significant association between death and sex (aOR = 2.4; 95%CI = 1.3–4.4), age (aOR = 1.08; 95%CI = 1.05–1.11), and chronic disease (aOR = 3.5; 95%CI = 1.3–9.6). When the same analysis was limited to patients 65+ years of age, there was an association between death and sex (aOR = 2.3; 95%CI = 1.2–4.6), age (aOR = 1.07; 95%CI = 1.03–1.12) and chronic disease (aOR = 3.53; 95%CI = 1.01–12.33). In neither analysis was influenza vaccination associated with the outcome (*p* > 0.05; Table 2), regardless of the vaccine formulation (*p* > 0.05; not shown).

## 4. Discussion

Our study found no association between influenza vaccination and disease severity or risk of death in COVID-19 patients. This result is in accordance with a 2020 Italian study [19] of 17,600 inhabitants from the Italian province of Reggio Emilia who were tested for SARS-CoV-2. The authors concluded that influenza vaccination did not affect either hospitalization or death in COVID-19 patients, although it was associated with a lower probability of a COVID-19 diagnosis (not evaluated in our study). A US study [20] enrolled 1434 patients with COVID-19 at the Cleveland Clinics in Ohio and Florida between March 8 and April 15, 2020. The multivariate analysis showed no impact of influenza vaccination on the risk of hospitalization (aOR = 1.29; 95%CI = 0.72–2.31) or in-hospital mortality (aOR = 0.98; 95%CI = 0.39–2.43), thus demonstrating that vaccination had no effect on morbidity or mortality related to COVID-19.

Our study instead showed that sex, age, and chronic disease are the main risk factors for COVD-19-related hospitalization and death. Specifically, it identified males as being at higher risk of morbidity and mortality from COVID-19. Sex-based differences in the immune response to SARS-CoV-2 were reported in a 2021 paper published in *Science* [21], which attributed the higher risk of severe disease and death from COVID-19 in males to differences in male vs. female physiology with respect to innate immune cytokines and chemokines (interleukins-8 and 18), type I interferon, T cell activation, sex chromosomes, immune cell transcriptomes, cytokine responses to viral infections, and sex hormones. The increased risk of COVID-19-related complications and death as determined in the multivariate regression models is depicted in Figure 1. The greater risk of increased COVID-19 morbidity and death in older patients is well established. According to the CDC, compared to patients aged 18–29 years old, the risk of hospitalization in older age groups is 8- to 13-fold higher and the risk of death is 90- to 630-fold higher [22]. The CDC also identified chronic diseases as risk factors for morbidity and mortality in COVID-19 patients [22], in agreement with our results.

Our study is one of the few to examine the potential impact of the influenza vaccine on COVID-19. The study’s design allowed us to estimate the cause–effect relationship between the vaccine and COVID-19 severity (even if more evidence is needed to clarify this topic). Integration of the dataset obtained from the GIAVA COVID-19 surveillance platform with supplementary information collected from other archives resulted in data that had been verified and were thus highly accurate. However, it should also be noted that the referenced archives were established for administrative and non-epidemiological purposes such that, at least theoretically, there was a risk of related bias. Future studies should similarly evaluate the role of other vaccines (e.g., anti-pneumococcal vaccine) and different chronic diseases (e.g., heart disease, diabetes, respiratory diseases, etc.) as risk factors in COVID-19. A major limitation of the study was that the analysis could not account for the effect of the strict nationwide lockdown (9 March to 3 May 2020), which altered the natural course of the disease by limiting its spread, especially in certain settings (hospitals, care home for the elderly, essential businesses, etc.). Moreover, surveillance activities during the first wave of the pandemic were specifically focused on hospitals and nursing homes, given their high-risk populations. This strategy may have distorted the general picture of the disease’s epidemiology and delayed or hindered the detection of asymptomatic or pauci-symptomatic COVID-19 patients, who were more likely to have been younger and otherwise in good health. Finally, the sample size was limited, but it must be considered that the province of Brindisi is the smallest province of Apulia region.

Although influenza vaccination has no impact (either positive or negative) on the severe manifestations of COVID-19, it must be promoted as an important public health measure during the coronavirus pandemic, given that, as noted by Paget J and Caini S [23], by preventing influenza infections it can both simplify the differential diagnosis of COVID-19 and free up healthcare facilities to deal with these patients. Indeed, one of the major issues of the COVID-19 pandemic has been the overwhelmed healthcare systems [24]. Furthermore, influenza vaccination has an essential role in protecting older adults, with higher coverage among the elderly, vulnerable individuals with chronic conditions, healthcare workers, and workers in other essential services; it may also influence the health effects and social consequences of the COVID-19 pandemic [25,26]. A topic of further interest is vaccine coverage during the 2020–2021 influenza season compared with previous seasons, including whether, despite the pandemic, vaccination against influenza remained a priority among public health institutions, the general population, and especially higher risk groups.

## 5. Conclusions

In summary, during this period in which mass anti-COVID-19 vaccination campaigns have yet to be implemented in many countries, influenza vaccination can serve as a supportive measure in the management of the pandemic and reduce the burden on patients and healthcare systems.

## Figures and Tables

**Figure 1 vaccines-09-00358-f001:**
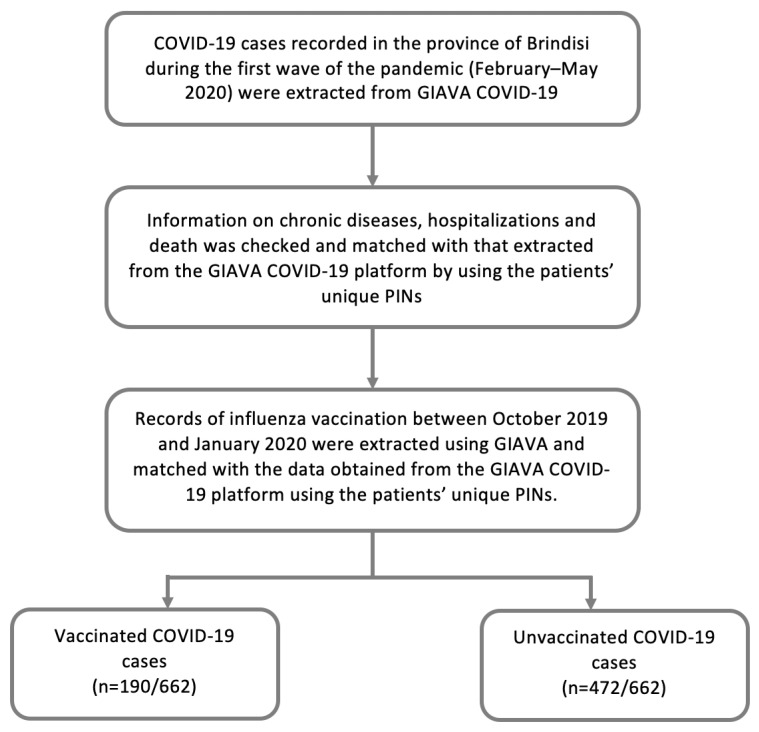
Study population.

**Figure 2 vaccines-09-00358-f002:**
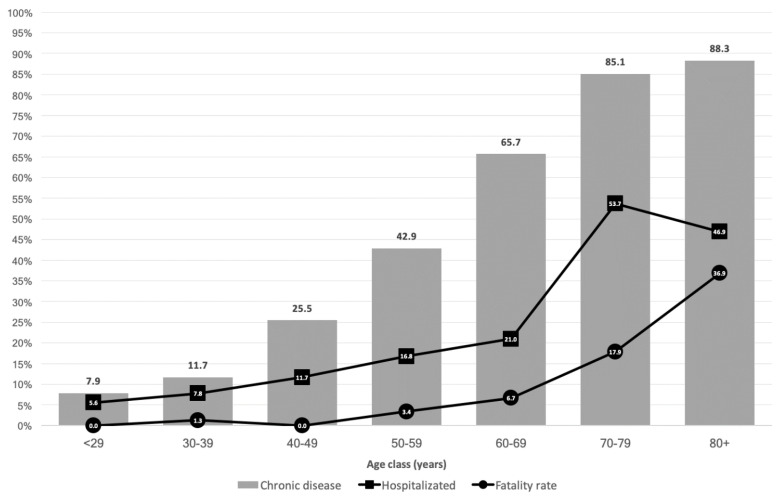
Proportion of patients per age class with at least one chronic disease, hospitalization rate for COVID-19, and fatality rate.

**Table 1 vaccines-09-00358-t001:** The determinants of hospitalization according to multivariate logistic regression models for the total population and for patients 65+ years of age.

Determinant	Total Population			65+ Years		
	aOR	95%CI	*p*-Value	aOR	95%CI	*p*-Value
Influenza vaccine (YES/NO)	1.2	0.7–1.9	0.510	1.03	0.56–1.92	0.914
Sex (male vs. female)	2.3	1.5–3.5	<0.0001	1.78	0.99–3.19	0.053
Age (years)	1.03	1.02–1.05	<0.0001	1.02	0.99–1.06	0.159
Chronic disease (YES/NO)	2.7	1.6–4.4	<0.0001	7.8	2.6–23.3	<0.0001
Goodness-of-fit	Hosmer-Lemeshow chi-squared = 3.6; *p* = 0.892			Hosmer-Lemeshow chi-squared = 6.1; *p* = 0.637		

**Table 2 vaccines-09-00358-t002:** The determinants of death according to multivariate logistic regression models for the total population and patients 65+ years of age.

Determinant	Total Population			65+ Years		
	aOR	95%CI	*p*-Value	aOR	95%CI	*p*-Value
Influenza vaccine (YES/NO)	1.6	0.8–3.2	0.165	1.7	0.8–3.6	0.162
Sex (male vs. female)	2.4	1.3–4.4	0.007	2.3	1.2–4.6	0.016
Age (years)	1.08	1.05–1.11	<0.0001	1.07	1.03–1.12	<0.0001
Chronic disease (YES/NO)	3.5	1.3–9.6	0.013	3.53	1.01–12.33	0.048
Goodness-of-fit	Hosmer-Lemeshow chi-squared = 8.2; *p* = 0.417			Hosmer-Lemeshow chi-squared = 4.1; *p* = 0.845		

## Data Availability

Data available on request due to restrictions eg privacy or ethical.

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
