# Peer review of "Influenza Vaccination and Health Outcomes in COVID-19 Patients: A Retrospective Cohort Study"

_vaccines, 2021, doi:10.3390/vaccines9040358_

Round 1

Reviewer 1 Report

This manuscript looks for the potential association of influenza vaccination with COVIID-19 hospitalization and death with no such association being observed. Following changes are suggested for the authors to make the manuscript clearer:

  1. In introduction, in the sentences 'The Who reported ...........EU/EEA', the numbers keep changing over time. Authors need to provide date of which the numbers are reported.
  2. Include a flow diagram showing the study population: influenza vaccinated vs non-vaccinated SARS-CoV-2 positive cases. 
  3. Authors mentioned inclusion of trivalent and tetravalent vaccines. Were there any difference in terms of which vaccine was used and COVID-19 outcome. 
  4. Table 2 legends should be for death not hospitalizations. 
  5. In discussion, the third paragraph 'In summary,..........to the disease' is repeated. Remove it. 

Author Response

A1. Revised

A2. Added.

A3. As reported in methods section, a sub-analysis was performed for type of vaccine. Results of this sub-analysis did not show any difference per type of vaccine (we reported this sentence in results section).

A4. Revised.

A5. Revised.

Reviewer 2 Report

Dear Authors, this is an interesting paper in which the epidemiological data collected at local level are adequately valorised. It would be useful to extend the study to other areas. I have only two minor suggestions:

  • In the discussion section, the sentence “ The study’s design allowed us to evaluate the cause-effect relationship between the vaccine and COVID-19 severity” should be amended. The assessment of a cause-effect relationship is a complex process that that is more than just demonstrating a statistical association.
  • The limited sample size should be mentioned among the study's limitations.

Author Response

A1. Revised.

A2. Revised.